# A New Approach to Low-Cost, Solar Salt-Resistant Structural Materials for Concentrating Solar Power (CSP) and Thermal Energy Storage (TES)

Fadoua Aarab [1], Bernd Kuhn [1,*], Alexander Bonk [2] and Thomas Bauer [2]

[1] Institute of Energy and Climate Research (IEK), Microstructure and Properties of Materials (IEK-2), Forschungszentrum Jülich GmbH, 52425 Jülich, Germany; f.aarab@fz-juelich.de
[2] German Aerospace Center (DLR), Institute of Engineering Thermodynamics, Linder Höhe Bldg. 26, 51147 Cologne, Germany; Alexander.Bonk@dlr.de (A.B.); Thomas.Bauer@dlr.de (T.B.)
* Correspondence: b.kuhn@fz-juelich.de; Tel.: +49-2461-61-4132

**Abstract:** "Concentrated solar power" (CSP) and thermal energy storage (TES) are promising renewable energy technologies, which have gained increasing interest and practical application in recent years. CSP and TES systems typically utilize molten salts such as the so-called "solar salt", a mixture of 60 wt.% $NaNO_3$ and 40 wt.% $KNO_3$, for heat transfer and storage. The overall efficiency of commercially operating CSP and TES systems is currently limited, because of solar salt thermal stability, which prevents process temperatures higher than 600 °C. Even at these temperatures, corrosion of the structural materials applied in salt guiding pipework, tubes and containers is a matter of concern in long-term operation, which necessitates careful material selection. This paper outlines the superior salt corrosion behavior of a novel low-cost, $Al_2O_3$-forming, ferritic, Laves phase-strengthened (i.e., structural) steel in $NaNO_3/KNO_3$ solar salt at 600 °C. Directions for the further development of the LB2230 trial steel towards improved structural properties are derived in comparison to its predecessor Crofer®22 H.

**Keywords:** concentrating solar power; salt corrosion resistance; protective $Al_2O_3$ scale; structural material development

## 1. Introduction

The demand for clean, renewable, efficient and affordable energy technologies rises due to the limited supply of fossil resources and the impact of $CO_2$ emissions on global warming. In combination with thermal energy storage (TES), efficiency [1] and capacity make concentrating solar power (CSP) plants a very promising solution for the provision of uninterrupted base-load electricity supply. So-called Carnot batteries or storage plants [2–7] for the substitution of fossil-fired boilers by TES systems and further utilization of the power blocks of existing steam power plants provide a cost-effective way of grid stabilization and thus a key element in transitioning the energy system [2]. In CSP/TES power plants, a heat transfer fluid is heated by concentrated solar energy (and stored when combined with TES), whereby electrical energy is generated in a downstream steam power process. The use of molten salt as the heat transfer and storage medium has already been demonstrated in a number of projects [8,9]. Gemasolar (located in Spain, in operation since 2008), the first commercial solar power plant, utilizes solar salt, a eutectic mixture of 60 wt.% $NaNO_3$ and 40 wt.% $KNO_3$, for heat transfer and storage [10]. A disadvantage of using molten nitrates in thermal energy storage devices is their high corrosiveness towards metal piping and container walls [11,12]. Salt corrosion issues in long-term operation of about 30 years [12] currently necessitate the application of comparatively expensive Ni-base super alloys or austenitic stainless-steel grades [13]. For this reason, investment, and maintenance costs, involved in the usage of these high-price structural materials, are a major price driver and one of the main reasons for lacking competitiveness of CSP technology [14].

Even in the sunny regions of the world, such as southern Europe, North and South Africa or Nevada, electricity generation from CSP currently lacks economic feasibility in competition with conventional or photovoltaic plants [15]. Experts agree that rising prices for fossil fuels will make CSP competitive in the future [15], but the thermal efficiency of CSP plants currently ranges from 30% to 40%. To increase the competitive ability, a rise in thermal efficiency up to the level of the most advanced steam power plants of 46% [16] is necessary, but requires higher working temperature of the salt, which in turn is related to increased corrosion issues [17].

Most of the literature on corrosion of high-temperature structural alloys in molten nitrate salt deals with $Cr_2O_3$-forming, low-alloy steels, austenitic stainless-steels or Ni-base super alloys [8,18–23]. A high Ni content of the alloy matrix obviously helps in reducing the weight gain in isothermal corrosion experiments. Nevertheless, chromia layers formed on component surfaces cannot provide sufficient long-term protection, because of their solubility in nitrate salt [24]. Few studies are available on $Al_2O_3$-formers, which exhibit superior resistance to corrosion in molten salt [21,25]. Investigation into and development of sufficient salt corrosion, heat resistant and at the same time low-cost structural materials for this reason provides a promising lever for cost reduction and thus the market breakthrough of CSP technology. This paper outlines the performance of a new type of Laves phase-strengthened, low-cost, salt corrosion-resistant ferritic steel (called "LB2230", cf. Section 2)

## 2. Materials and Methods

### 2.1. Materials and Preparation

2.1.1. Structural Materials

In the context of this work, two ferritic stainless-steels, strengthened by Laves phase precipitates, were investigated: Crofer®22 H, a chromia former, and LB2230, an experimental alumina-forming derivate of Crofer®22 H. The high chromium content of Crofer®22 H provides excellent steam oxidation and corrosion resistance [26]. The high creep resistance of this type of steel is achieved by combined solid solution and intermetallic Laves particle precipitation strengthening [26]. This in principle makes the steel conceivable for structural CSP power plant application, with long-term resistance to molten salt corrosion remaining a point of concern. LB2230, an $\alpha$-$Al_2O_3$-forming, Laves phase-strengthened derivate of Crofer®22 H, potentially provides significantly better molten salt corrosion resistance compared to its chromia-forming forerunner Crofer®22 H [27].

The performance of these two low-cost ferritic grades is compared to the alumina-forming Ni-base super alloy Haynes 214, which was developed for application in molten salt and for this reason serves as a benchmark.

Crofer®22 H

Crofer®22 H was originally developed for application in high-temperature fuel cell systems. The Cr content of 22 wt.% provides high steam oxidation resistance and thermal expansion closely matched to the ceramic reaction layers of the fuel cell. By additions of W, Nb and Si, increased mechanical properties, mainly improved creep resistance [26,28–30], are achieved through solid solution hardening and precipitation strengthening by intermetallic Laves phase particles. At temperatures up to 900 °C, an oxide bi-layer consisting of Cr-Mn spinel forms on top of a thermodynamically stable (in air and $H_2/H_2O/CO/CO_2/N_2$ mixtures) chromia layer [30–34]. The chemical composition of Crofer®22 H is presented in Table 1. The thermal expansion coefficient of Crofer®22 H between 20 and 600 °C is 11.2 ppmK$^{-1}$ [30].

LB2230

LB2230 is a derivate of Crofer®22 H and was developed and patented by Forschungszentrum Jülich GmbH and VDM Metals GmbH under funding from the German Federal Ministry of Education and Research (BMBF) within the WING (Werkstoffinnovation fuer Indus-

trie und Gesellschaft) framework project "Ferrit 950" (grant number: 03x3520E). At temperatures beyond 1000 °C, a protective $\alpha$-$Al_2O_3$ top layer is formed in air. Like Crofer®22 H, LB2230 contains tungsten, niobium and Si for combined solid solution and precipitation strengthening. The effect of Nb in LB2230 is two-fold: it is a key element in Laves phase precipitation [26,35,36] and it prevents the formation of chromium carbides at grain boundaries, which can lead to intergranular corrosion [37].

The addition of tungsten ensures solid solution hardening and increases Laves phase volume fraction [36]. The intermetallic (Fe,Cr,Si,Al)2(Nb,W/Mo) Laves phase particles considerably increase the mechanical strength of the alloy [38,39]. LB2230-like steel for this reason provides a unique combination of low cost, ease of fabrication and enhanced mechanical strength over commercial FeCrAl alloys already successfully tested for corrosion resistance against molten salts [25,40,41]. The addition of silicon promotes Laves phase formation, stabilizes it [42,43] and increases the service life of Fe-Cr steel components by improved adhesion of the protective oxide layers [44]. The thermal expansion coefficient of LB2230 between 20 and 600 °C is 14.4 ppmK$^{-1}$. A typical composition of this type of steel is presented in Table 1.

Haynes 214

In air, the nickel-base alloy Haynes 214 (chemical composition presented in Table 1) forms a protective $Al_2O_3$ surface layer at 955 °C and above [45], while at temperatures below 955 °C, a mixed chromia/alumina layer of lower corrosion resistance [27,45] develops. In [46], the use of Haynes 214 in molten salts is suggested up to a temperature of 850 °C. At such temperatures (ranging from 595 to 925 °C), it will age-harden by precipitation of $\gamma'$ ($Ni_3Al$) particles and consequently be susceptible to strain-age cracking [44]. The thermal expansion coefficient of Haynes 214 between 20 and 600 °C is 15.2 ppmK$^{-1}$ [45].

**Table 1.** Chemical compositions of the investigated materials (wt.%).

| Material | C + N | Cr | Ni | W | Nb | Si | Al | Mn | Ti | La | Zr | Fe |
|---|---|---|---|---|---|---|---|---|---|---|---|---|
| Crofer®22 H | 0.02 | 22.94 | - | 1.94 | 0.51 | 0.2 | - | 0.43 | 0.07 | 0.08 | - | R |
| LB2230 | <0.01 | 18.9 | - | 2.0 | 0.46 | 0.28 | 3.4 | 0.28 | 0.01 | - | - | R |
| Haynes 214 | 0.05 | 16.0 | 75 | - | - | 0.2 | 4.5 | 0.5 | 0.01 | - | 0.14 | R |

Three sheet specimens of 10 mm × 10 mm × 1 mm in dimension were cut from each of the alloys. According to ISO 17245:2015 [47], the surfaces and edges of these were ground by grit 600 SiC paper to ensure comparable intial surface roughness conditions. Finally, the samples were cleaned with ethanol and dried by hot air.

In air, the formation of protective $\alpha$-$Al_2O_3$ scales on metal surfaces typically necessitates temperatures of 1000 °C or higher [48]. For this reason, LB2230 and Haynes 214 were pre-treated to be corrosion tested in the pre-oxidized state as well. For this purpose, annealing at 1000 and 1100 °C was performed on Haynes 214 and at 1075 and 1100 °C on LB2230 in laboratory air for 1 h. In case of Haynes 214, the temperature levels were chosen according to data sheet [45] information, and in case of LB2230, they were chosen to prevent uncontrolled precipitation of Laves phase particles, which can take place below 1050 °C [39].

2.1.2. Solar Salt

For this study, high-purity salts (specifications presented in Table 2) were used to avoid impurity effects as much as possible. A mixture of 60 wt.% $NaNO_3$ (supplier: VWR Chemicals, Radnor, PA, USA) and 40 wt.% $KNO_3$ (supplier: Merck KGaA, Darmstadt, Germany) was prepared.

**Table 2.** Salt specifications (parts per million (ppm), unless otherwise stated).

| Material | Purity (%) | pH (25 °C) | Heavy Metals | Cl | $IO_3$ | $NH_4$ | $PO_4$ | $SO_4$ | Ca | Fe | Mg | Cu | Na |
|---|---|---|---|---|---|---|---|---|---|---|---|---|---|
| $NaNO_3$ | ≥99.5 | 5.5–8.3 | ≤5 | ≤5 | ≤10 | ≤20 | ≤5 | ≤30 | ≤20 | ≤3 | ≤20 | - | - |
| $KNO_3$ | ≥99.0 | 5.0–7.5 | ≤5 | ≤5 | ≤5 | ≤10 | ≤5 | ≤30 | ≤10 | ≤3 | ≤15 | ≤1 | ≤200 |

The test crucibles were filled with salt just below the upper rim and the respective metallic samples were placed inside. Weighing was performed separately for each test crucible. The salts were not pre-treated, unless stated otherwise.

### 2.2. Experimental Methods

#### 2.2.1. Corrosion Testing

The corrosion experiments were carried out by two different methods in two differing facilities. According to ISO 17245:2015 [47], a horizontal alumina tube furnace (Carbolite Gero GmbH, Neuhausen, Germany) was utilized for discontinous corrosion testing at Forschungszentrum Jülich (FZ Jülich), IEK-2. A schematic of the experimental set-up is depicted in Figure 1.

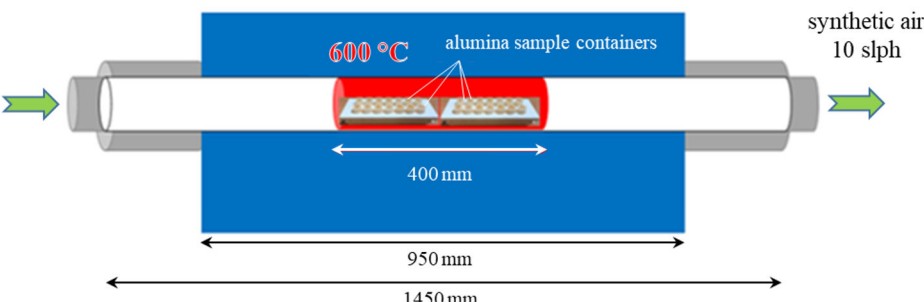

**Figure 1.** Schematic of the salt corrosion experimental set-up utilized at Forschungszentrum Jülich GmbH.

Two specimens of each alloy were placed in individual alumina sample containers, which were filled with salt just below the upper edge. Discontinous annealing was carried out for a total of 2042 h at 600 °C in synthetic air (flow rate: 10 sl/h), flushed through the tube furnace. The samples were weighed to determine their individual mass changes and the salt levels were checked after certain intervals. For this purpose, the containers were removed from the hot furnace, and the samples were taken out of the liquid salt and cooled in air. Salt residues were removed by rinsing with warm, distilled water. Subsequently, the samples were cleaned in ethanol and dried in hot air.

At the German Aerospace Center (DLR), the third specimen of each alloy was isothermally aged in the same salt composition within an autoclave set-up [49,50] under a continuous flow of synthetic air of 6 sl/h. The samples were all placed in a common alumina crucible (i.e., no individual crucible for each specimen) and then covered by the salt mixture. The crucible was then placed into a tubular, heated stainless-steel chamber. The continous salt flow over component surfaces in solar thermal power plants, which may produce different oxide morphologies, was simulated by continously stirring the salt during the experiment. Due to isothermal aging, the sample masses were determined before and after termination of the experiment only.

The area-specific mass changes of the samples were calculated according to DIN 50905-1 [51] by:

$$m_M = \frac{\Delta m_m}{A_i} = \frac{m_n - m_i}{A_i}, \tag{1}$$

where $\Delta m_m$ symbolizes the change in mass between the initial weight, $m_i$, and the respective weight measurement, $m_n$, after the given time intervals. $A_i$ represents the total sample surface at the beginning of the experiment.

### 2.2.2. Chemical Salt Analysis

The salt compositions were analyzed (by both Forschungszentrum Jülich, ZEA-3, and DLR, Institute for Technical Thermodynamics) before and after termination of the corrosion experiments carried out at FZ Jülich. After the experiments, the salts were poured from the containers onto ceramic plates for rapid cooling and subsequently ground. Quantitative analysis for $NO_3^-$ and $NO_2^-$ was carried out by diluting 75 mg of salt in 50 mL of milli-Q water each and applying a Thermo Fisher (Thermo Fisher Scientific, Schwerte, Germany) ICS-3000 ion chromatograph, equipped with an AS 14A Thermo Fisher analysis column, using a 1 mM $NaHCO_3$/8 mN $Na_2CO_3$ eluent at FZ Jülich, ZEA-3. Ions were detected by suppressed conductivity detection using a Metrohm suppressor module (MSM). Independent double-analysis at DLR was accomplished by dissolving 2 mg of salt each in 100 mL of ultrapure water (HiPerSolv, VWR, Darmstadt, Germany), utilizing a Metrohm Compact IC Flex 930 (Metrohm, Herisau, Switzerland) ion chromatograph, equipped with a Metrosep A Supp analysis column. Ions were detected by suppressed conductivity detection with dilute sulfuric acid (0.1M TitriPUR®, Merck Millipore, Darmstadt, Germany) as the suppressor. Na, K and the main alloying elements of the exposed metal samples (i.e., Cr, Fe, Ni, Nb, W and Al) were analyzed utilizing a Thermo Scientific iCAP 6500 atomic emission spectrometer with inductively coupled plasma. For this purpose, two 50 mg samples of each salt were dissolved in 3 mL of HCl/2 mL of $HNO_3$/1 mL of $H_2O_2$ each and replenished to 50 mL total volume. Two aliquots of each salt were analyzed in 1/100 and 1/10 dilution.

### 2.2.3. Metallographic Preparation and Microstructural Investigation

After corrosion testing, the samples were removed from the salt-filled containers, cleaned, sputtered with gold and electroplated by a thin layer of nickel to prevent the oxide layer(s) at the surface from cracking during metallographic preparation. After this, the samples were cold-mounted in epoxy resin, ground and polished applying diamond polish solution down to <1 μm of surface roughness and characterized by scanning electron microscopy with energy and/or wavelength dispersive X-ray spectroscopy (SEM/EDX— Zeiss Supra 50VP/Oxford Instruments Inca/Wave). SEM images of higher resolution (4096 × 3072 pixels) were taken of Crofer®22 H and LB2230 for the analysis of precipitate microstructure. The images were analyzed quantitatively (utilizing the software package OLYMPUS Stream Desktop) applying the method reported and succesfully applied to Laves phase-strenghtened alloys by Lopez et al. [42,43,52,53].

## 3. Results and Discussion

### *3.1. Pre-Oxidation*

In air at temperatures higher than 1000 °C, austenitic Haynes 214 and ferritic LB2230 form protective $Al_2O_3$ scales, while mixed chromia/alumina layers will form below this temperature [38,45]. The addition of aluminum forms a tightly adhering $Al_2O_3$ layer, which protects against chromium vaporization. For this purpose, additions of >3 Ma.-% Al are necessary in ferretic steels [54].

Information on corresponding temperature limits for the formation of protective $\alpha$-$Al_2O_3$ in solar salt is neither available for Haynes 214 nor LB2230. For this reason, both materials were corrosion tested in the (air) pre-oxidized condition, too. Figure 2 displays micrographs taken from the oxide layers formed by air pre-oxidation treatment. A continous alumina layer was not achieved by 1000 °C/1 h pre-oxidation in case of Haynes 214. Instead, a compact chromia layer formed on top of a discontinous, patchy alumina layer (Figure 2a) at the metal interface. In order to achieve a more uniform oxide layer, the pre-oxidation temperature was increased to 1100 °C, which resulted in a mostly continous alumina layer at the metal interface. A partly continous chromia layer was formed on top of this alumina scale, while at other locations the inhomogenous chromia layer still prevailed (Figure 2b).

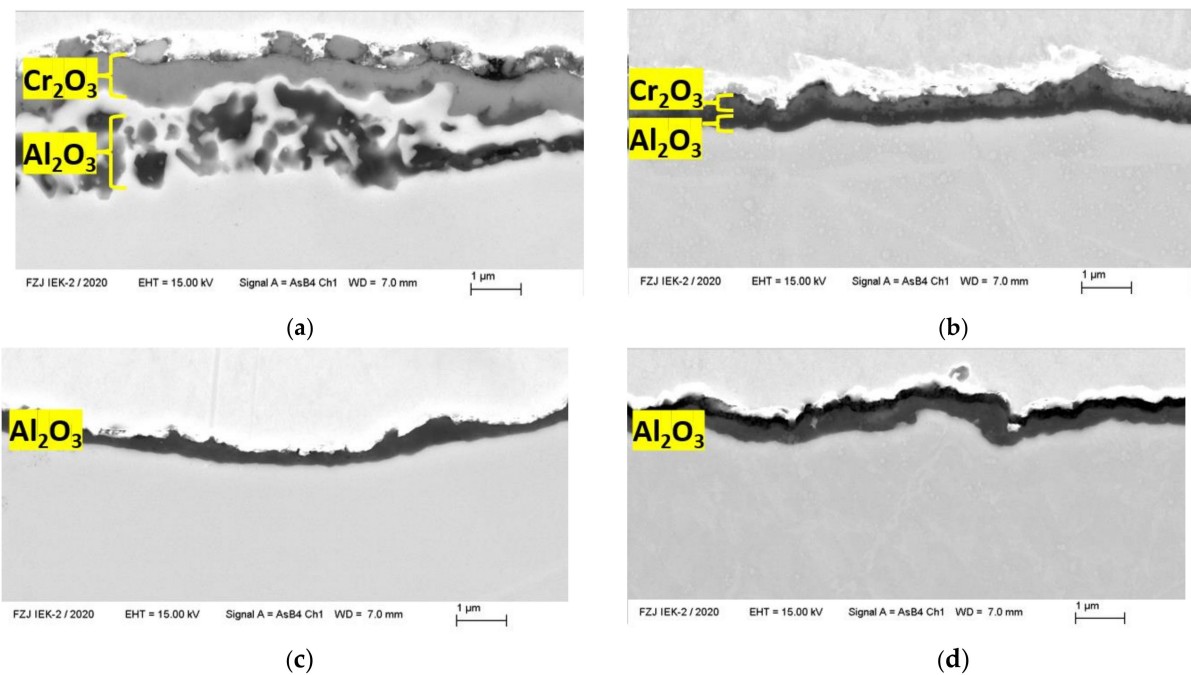

**Figure 2.** Oxide layers formed after 1 h of pre-oxidation treatment in air: Haynes 214 at (**a**) 1000 °C and (**b**) 1100 °C, LB2230 at (**c**) 1075 °C and (**d**) 1100 °C.

On ferritic, aluminum-containing steels, stable alumina scales are typically formed at temperatures above 950 °C [55]. In case of LB2230, a temperature of 1075 °C is necessary to prevent uncontrolled precipitation of Laves phase particles. After 1 h of annealing, a continous alumina layer was achieved (Figure 2c). In comparison to Haynes 214, the rise in pre-oxidation temperature merely resulted in an increase of $Al_2O_3$ layer thickness from about 0.2 to 0.8 μm (Figure 2d).

### 3.2. Weight Change

At short exposure times (up to 254 h), Haynes 214 yielded an initial mass increase no matter whether it was pre-oxidized in air or not (cf. Figure 3a,b). The comparably high thermal expansion coefficient of the Ni-base alloy may then have led to spallation of the formed oxide scale, which was accompanied by comparably strong mass loss between 254 and 550 h (Figure 3a). After this period, the mass changes stabilized at low levels and remain around mean values of $0.51 \pm 0.13$ mgcm$^{-2}$ and $-0.11 \pm 0.22$ mgcm$^{-2}$ (pre-oxidized) up to 2042 h.

In case of the ferritic, alumina-forming LB2230 alloy, the initial mass gain ranged on a lower level. With or without pre-oxidation in air, the initial rise was followed by a moderate but continuous drop in mass (starting from 96 h without pre-oxidation, 254 h with pre-oxidation), until 550 h was reached. Afterwards, the mass changes slightly increased again until 1046 h, before dropping to comparably low stable mean values of $0.17 \pm 0.02$ mgcm$^{-2}$ (non-pre-oxidized) and $0.05 \pm 0.005$ mgcm$^{-2}$ (pre-oxidized) up to 2042 h. The $Cr_2O_3$-forming Crofer®22 H presented gain/loss values twice as high, before it entered a stage of continuous mass loss after 1046 h. Compared to steam oxidation, to which $Cr_2O_3$-forming ferritic alloys such as Crofer®22 H are highly resistant [26], the salt corrosion-induced weight gains reported here represent an increase of almost two orders of magnitude.

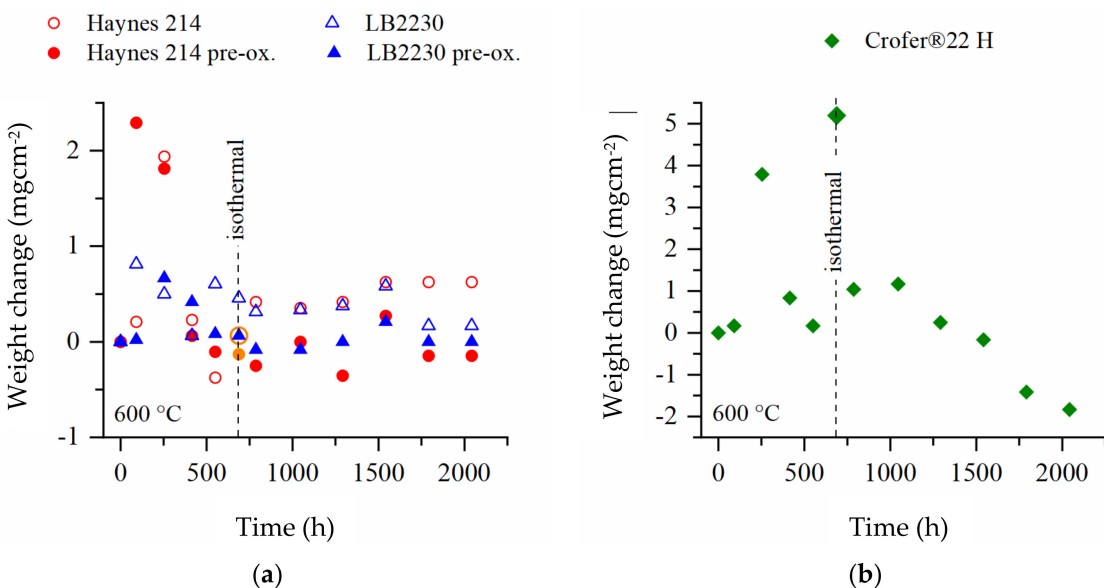

**Figure 3.** Weight changes of corrosion specimens during annealing in solar salt at 600 °C (discontinuously performed by FZ Jülich, IEK-2; isothermally by DLR): (**a**) $Al_2O_3$-formers and (**b**) $Cr_2O_3$-formers (mean values of 2 specimens in case of discontinuous testing, single value in case of continuous exposition).

The final mass changes obtained after 688 h of isothermal exposure in flowing salt at DLR correlate well to the trends measured in cyclic testing (cf. "isothermal" in Figure 3a) in case of the $Al_2O_3$-forming alloys. However, with 5.196 mgcm$^{-2}$ over a period of 688 h, the mass change of the $Cr_2O_3$-forming Crofer®22 H was approximately an order of magnitude higher than in discontinuous testing. None—not even Crofer®22 H—of the isothermally aged specimens presented visible spallation after the termination of the experiments. This indicates that in case of Crofer®22 H, spallation already played a major role, even in the early stages of cyclic testing (cf. Figure 4: bottom left).

Due to the still comparatively short exposure times and (micro) spallation, no dominant corrosion mechanism could be identified.

### 3.3. Microstructure

None of the materials displayed macroscopic indications of spallation, except Crofer®22 H. Scanning electron microscopy surface examination, presented in Figure 4, shows that the oxide on Haynes 214 formed in two layers: The bottom layer is fine-grained and consists of a mixed Na, Ni, Fe oxide. The top oxide layer, on the other hand, formed coarsely in a spot-like texture. On the pre-oxidized specimen, the oxide layer appears coarser and is furthermore characterized by bulges covering the entire area. The isothermally exposed specimen appears patchier than the cyclically exposed one. Superficial cracks appeared after 550 h, but not after 2042 h of exposure. The compositions of the oxide layers are largely similar, no matter what pre-oxidation state they are in. The only exception to this is an additional Al content in the pre-oxidation layer. Cross-sectional analysis with EDX sum spectra (Figure 5) demonstrates that the bottom oxide layer grows into the base material. Such internal oxidation typically indicates a non-continuous, or porous, non-protective $Al_2O_3$ layer. A protective $Al_2O_3$ layer normally should act as a diffusion barrier, so no internal oxidation should occur [56].

However, in experiments in [56] with exposure times of about 3000 h at 600 °C with air as the ullage gas, a somewhat protective mechanism of $Al_2O_3$ on Haynes 214 in solar salt was stated, which—at least in the long-term—seems doubtful in the light of the results outlined here.

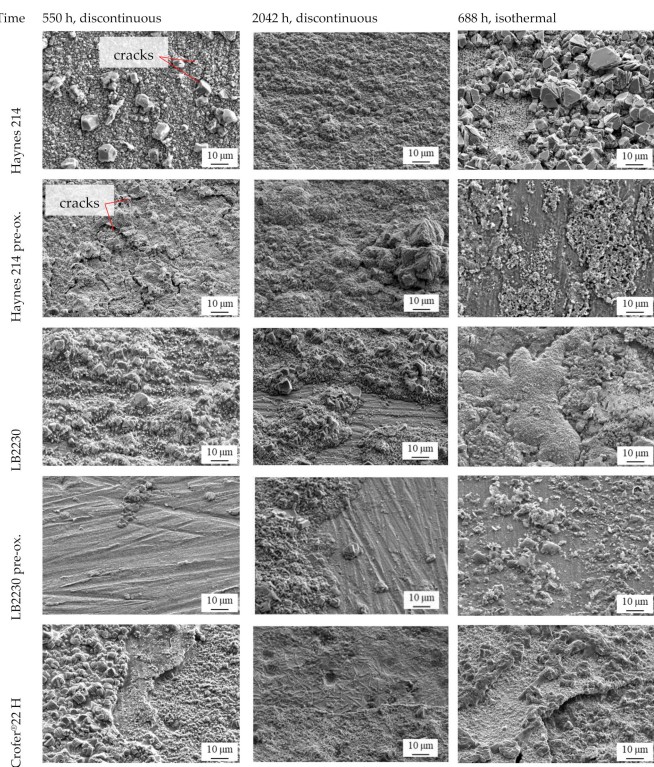

**Figure 4.** Scanning electron micrographs of the specimen surfaces after 550 h (**left**), 2042 h (**middle**) of discontinuous and 688 h of isothermal (**right**) corrosion testing in solar salt at 600 °C.

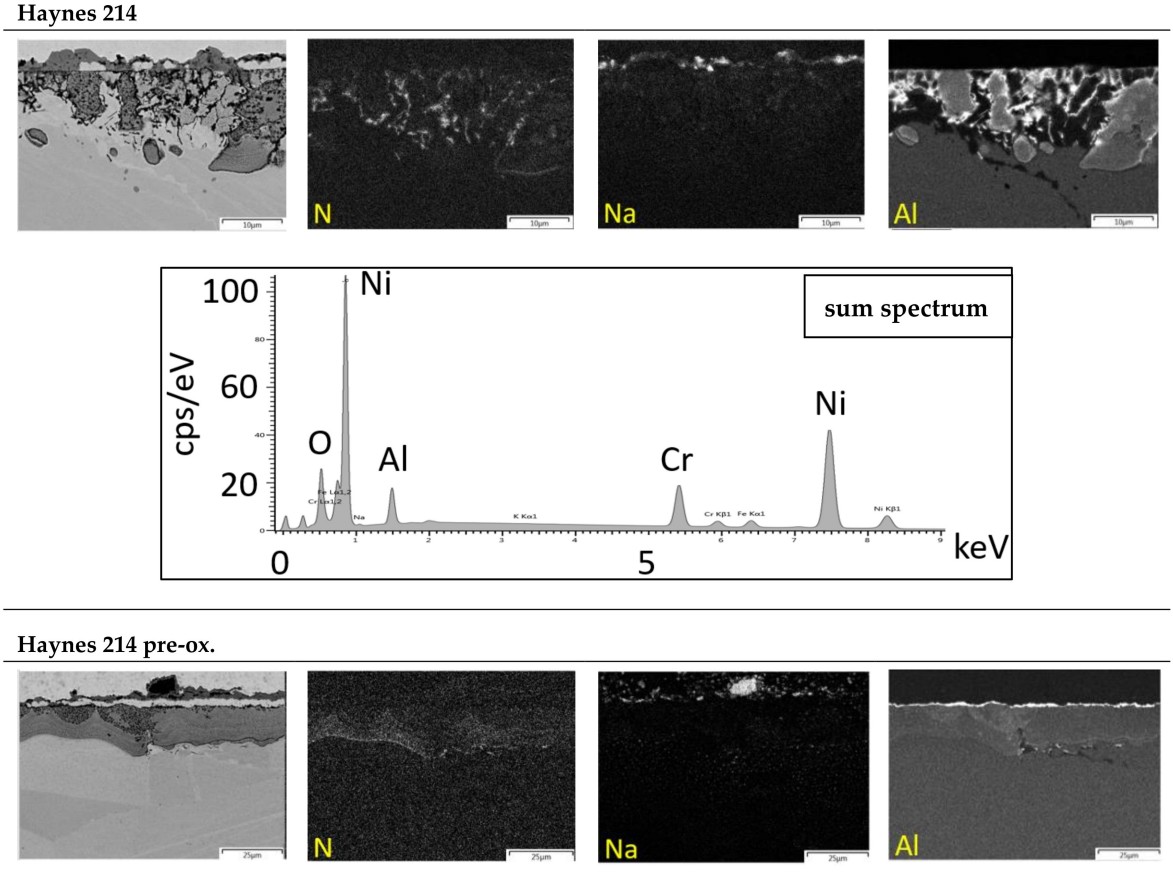

**Figure 5.** *Cont.*

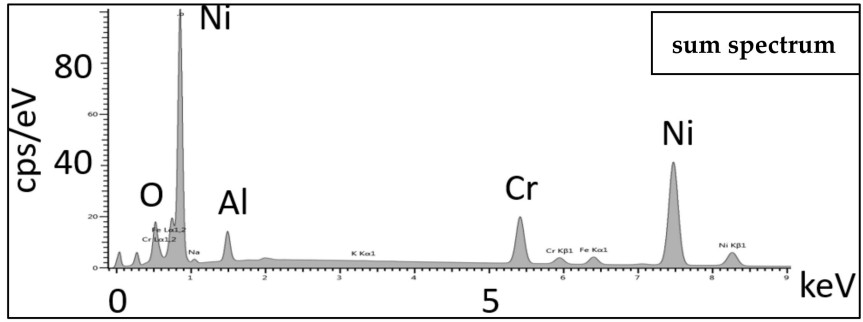

**LB2230**

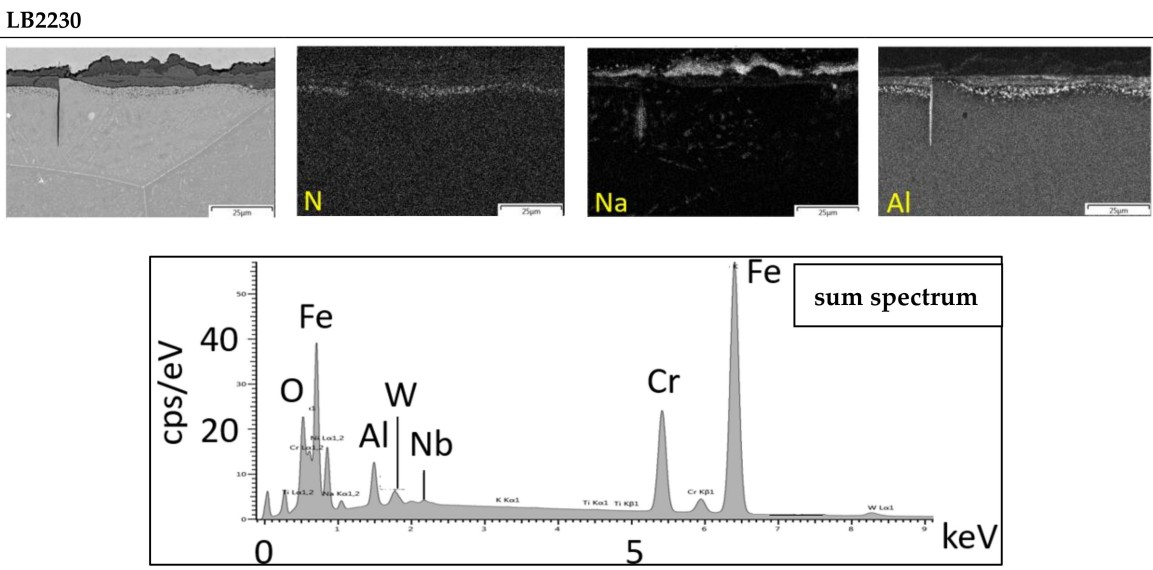

**LB2230 pre-ox.**

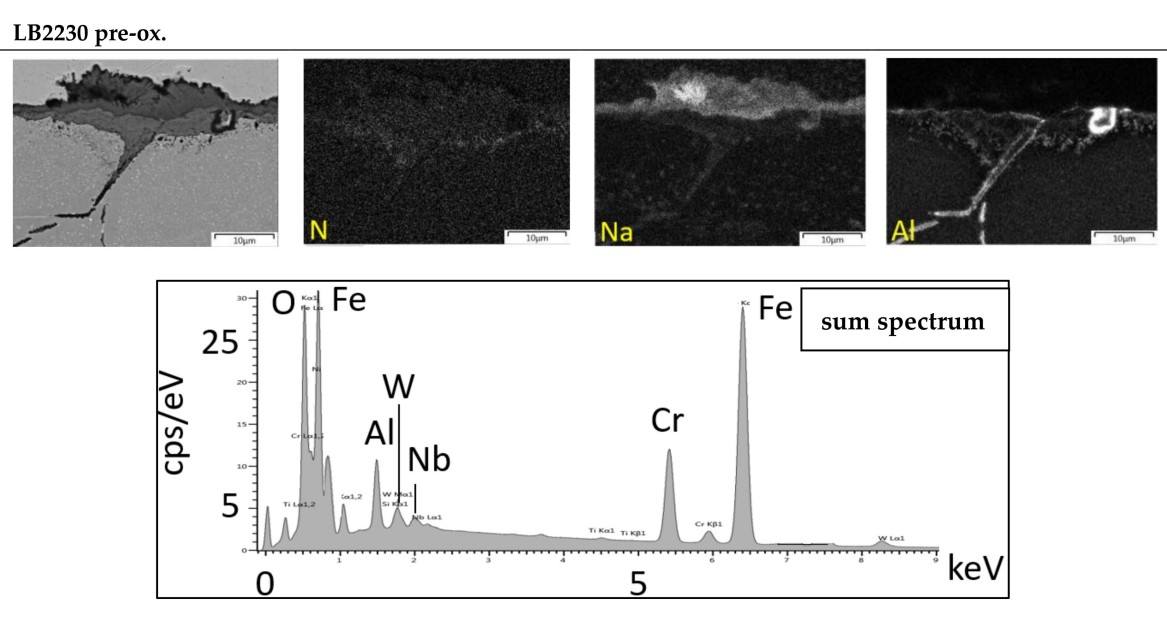

**Figure 5.** *Cont.*

**Crofer®22 H**

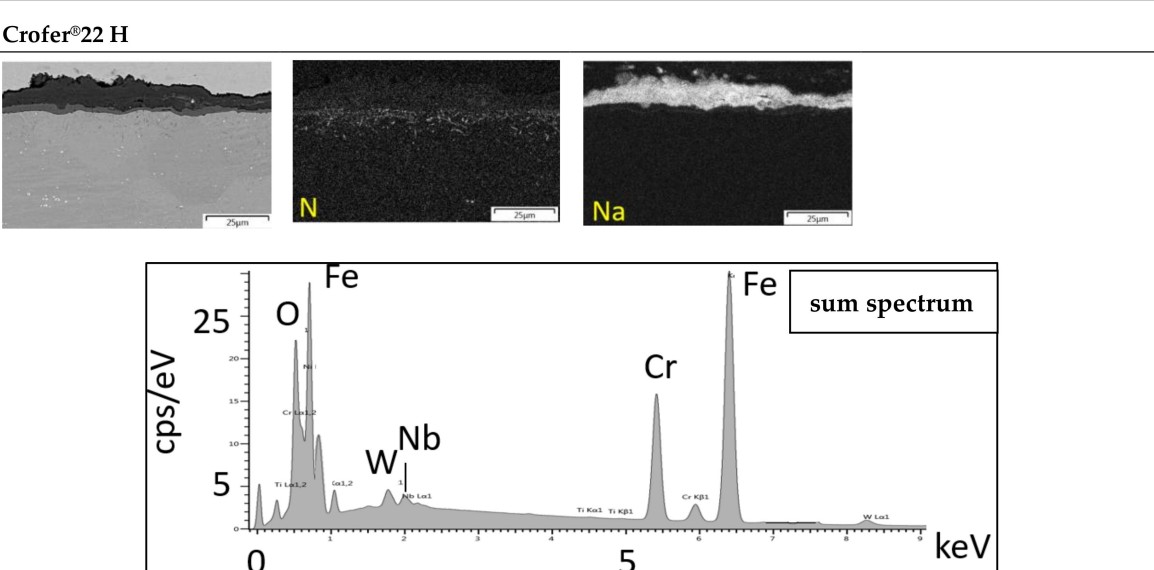

**Figure 5.** EDX element mappings of the sample cross-sections after 2042 h of discontinuous corrosion testing in solar salt at 600 °C with respective EDX sum spectra.

Scale cracking did not occur in case of LB2230, which is presumably caused by the lower mismatch of thermal expansion coefficients of the formed oxide scales and the ferritic matrix. An intermediate layer of Cr-Na-Fe-oxide formed between a top layer of Na-Fe mixed oxide and a protective bottom layer of $Al_2O_3$ on the steel substrate. In contrast to Haynes 214, internal oxidation of the LB2230 substrate material beneath the $Al_2O_3$ layer does not occur, which indicates higher stability of the protective layer. Intermetallic phase precipitates of Fe, Cr, Al, W and Nb form in the base metal (Figure 6), which are important for the structural properties of this kind of steel [44,57–60]. It is noteworthy that in solar salt, a protective $Al_2O_3$ layer is formed at the moderate temperature of 600 °C, even without pre-oxidation in air (which would not be possible by oxidation in air at such a low temperature). This means that LB2230-like steel is potentially self-passivating in exposition to solar salt.

On Crofer®22 H, the comparably thick oxide scales tended to crack and spalled during cooling due to stresses resulting from oxide growth itself and the difference in thermal expansion coefficients of the base metal and the oxides. Accordingly, the top layer of Na-Fe mixed oxide has an inferior protective effect against the corrosive salt bath. A thin Cr mixed oxide layer formed below the top layer. By avoiding temperature fluctuation (i.e., in isothermal aging), less spallation occurred, but the top oxide layer was less compact.

The element mappings of LB2230, Haynes 214 and pre-oxidized Haynes 214 (Figure 5) confirmed both Na mixed and N-containing phases within and below the oxide layer. Considering the $NO_2^-$ and $NO_3^-$ contents after exposure in comparison to the initial values (cf. Section 3.4), the formation of further, "nitrate-consuming" species can be assumed. K was not detected in any of the alloys.

Laves Phase Precipitation

Laves phases prove to be quite stable in the presented kind of steel [49,50]. The Laves phase precipitation is shown for LB2230, pre-oxidized LB2230 and Crofer®22H after 550 h of discontinuous corrosion testing in Figure 6. To provide a quantitative indication of the stability of the Laves phase particles in service, the mean size of the precipitates was analyzed after 550 and 2042 h of exposure in solar salt at 600 °C. Table 3 provides an overview of the mean particle diameter evolution in LB2230, pre-oxidized LB2230 and Crofer®22 H. Oxide scale growth and Laves phase precipitation interact in the near-surface areas, because Cr and Al take part in both oxide scale formation at the surface and precipitation within the metal matrix. For this reason, the particle population in near-

surface areas (down to 6 μm depth) must be analyzed separately from the one prevailing in the bulk material (depth of investigated area from 100 to 200 μm). The Laves phase particles appeared to be largely stable in the material bulk (i.e., almost constant particle diameter in the material bulk). Near the surface, considerably coarser particles appeared in LB2230 and pre-oxidized LB2230. Pre-oxidation of LB2230 interestingly led to an increase in size of the near-surface particles. In the case of Crofer®22 H, the size of the near-surface particles even decreased. $Al_2O_3$ is more stable than $Cr_2O_3$, which results in less consumption of Al and Cr by oxidation at the surface. Consequently, more Al and Cr is available for Laves phase precipitation in pre-oxidized LB2230. In the bare LB2230, the oxide layer forms during the aging process and consumes Al and Cr, which in turn hinders nucleation and growth of the Laves particles close to the surface (cf. Figure 6). With increasing material depth, the Al and Cr consumption at the surface no longer influences the Laves phase formation, so that the particle sizes equalize.

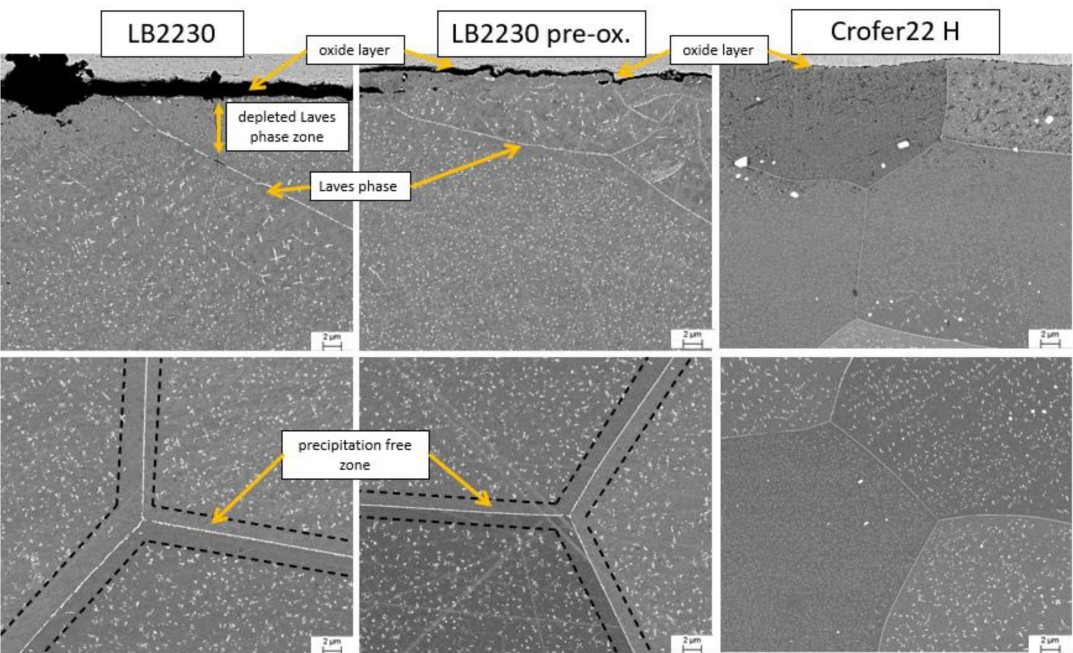

**Figure 6.** SEM images of Laves phases precipitation in LB2230, pre-oxidized LB2230 and Crofer®22 H after 550 h of discontinuous corrosion testing in solar salt at 600 °C.

**Table 3.** Mean equivalent circle diameters and standard deviation of Laves phase particles near the surface (down to 6 μm depth) and the material bulk (depth of investigated area from 100 to 200 μm) in LB2230, pre-oxidized LB2230 and Crofer®22 H.

| Material | Test Time | Near-Surface Area (nm) | Bulk Material (nm) |
|---|---|---|---|
| LB2230 | 550 h | 64.40 ± 20.36 | 115.47 ± 51.50 |
| | 2042 h | 123 ± 24 | 104.47 ± 49.54 |
| LB2230 pre-ox. | 550 h | 116.54 ± 53.61 | 116.09 ± 53.41 |
| | 2042 h | 152 ± 43 | 109.36 ± 55.85 |
| Crofer®22 H | 550 h | 82.09 ± 28.80 | 89.07 ± 43.45 |
| | 2042 h | 55.17 ± 12.05 | 77.56 ± 34.34 |

Due to the permanent Cr consumption at the surface of Crofer®22 H [61], the particle diameter diminishes towards the surface. The longer Crofer®22 H is aged, the more particles near the surface dissolve and the particle-depleted zone at the surface grows in depth. Homogenous precipitate microstructure, as in the case of LB2230, is desirable, because it provides maximized strength at the same material cross-section and thus enables thinner section components.

### 3.4. Salt Analysis

3.4.1. Nitrate Dissociation

Cr has high, while Fe and Ni have low solubility in nitrate salts [62]. Yellow-greenish discoloration of the salt, already apparent after 550 h of annealing at 600 °C (Figure 7), is typical and indicates dissolved chromates (e.g., (Na, K)$_2$CrO$_4$) [63], formed by oxidative dissolution of Cr$_2$O$_3$ [64].

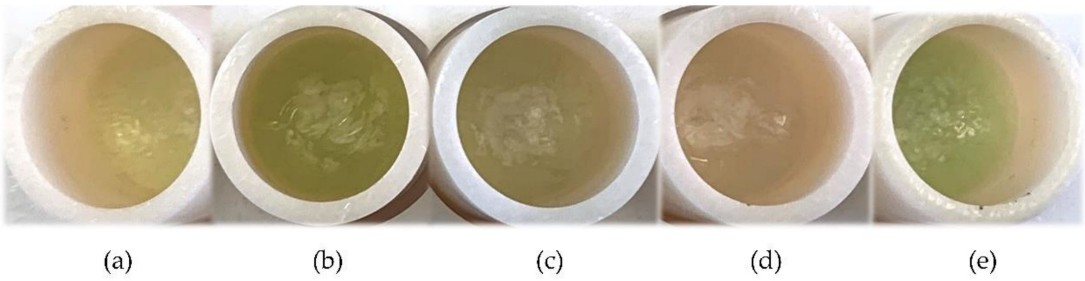

|   (a)   |   (b)   |   (c)   |   (d)   |   (e)   |

**Figure 7.** Discoloration of the solar salt in the Haynes214 (**a**), Haynes214 pre-ox. (**b**), LB2230 (**c**), LB2230 pre-ox. (**d**) and Crofer®22 H (**e**) containers after 550 h at 600 °C.

Table 4 lists the Na and K results of the salts taken from the containers after termination (550 and 2042 h) of the discontinuous corrosion experiments carried out at FZ Jülich, IEK-2. After 1542 h, the crucibles were refilled with fresh salt, i.e., the analyses were carried out at initially comparable salt samples, which were in service for 550 and 500 h, respectively, and for this reason, are well-comparable. Table 5 lists the NO$_3^-$ and NO$_2^-$ analysis (by FZ Jülich, ZEA-3) and the NO$_3^-$-, NO$_2^-$- and O$_2^-$-independent double-analysis (by DLR, Institute for Engineering Thermodynamics). The NO$_2^-$ content of the virgin salt was <0.01 mol% and increased during exposure from 0 to 550 h in all the containers, while the nitrate content decreased accordingly. As an example, the NO$_2^-$ content of the salt in the LB2230 container increased to 11.24 mol%. Simultaneously, the NO$_3^-$ content diminished from initially 99.99 mol% to 88.76/90.99 mol% from 0 to 550 h/1542 to 2042 h. The ratio between Na and K remains stable over time.

The corrosion layers, formed on all the samples, contained Na, while K was not detected in any case (cf. Figure 5). In solar salt itself, the equilibrium between NO$_2^-$ and NO$_3^-$ is reached within just a few hours at temperatures above 550 °C. At 600 °C, thermodynamic data by Nissen and Meeker indicate that the equilibrium nitrite content is 10.2 mol% [65]. The high nitrite contents of approximately 9.5–11.2 mol% (cf. Table 5) therefore indicate that the immersion of metallic samples is able to promote the dissociation of NO$_3^-$, which is supported by the presence of N-containing oxide and precipitate phases on and in the metal samples. The slight increase in O$_2^-$, caused by NO$_2^-$ dissociation, during exposure supports these results. More experimental data, especially concerning longer service times in combination with detailed microstructure and chemical analysis, are needed to elaborate valid corrosion mechanisms and to derive the potential impact on the mechanical properties of structural materials.

**Table 4.** Molar contents of Na and K in solar salt after 550 h and after 2042 h of discontinuous corrosion testing at 600 °C. Analysis by FZ Jülich, ZEA-3.

| Material in Salt | 550 Hours | | 1542–2042 Hours | |
|---|---|---|---|---|
|  | **Na** | **K** | **Na** | **K** |
| Solar salt (initial) | 65.33 | 34.67 | 65.33 | 34.67 |
| Haynes 214 | 63.60 | 36.40 | 62.73 | 37.27 |
| Haynes 214 pre-ox. | 64.92 | 34.08 | 78.95 | 21.05 |
| LB2230 | 63.60 | 36.40 | 61.16 | 38.84 |
| LB2230 pre-ox. | 63.60 | 36.40 | 61.16 | 38.84 |
| Crofer®22 H | 63.93 | 36.07 | 61.16 | 38.84 |

The molar fractions of nitrate and nitrite measured in both analyses are in good correlation (cf. Table 5). The nitrite contents after exposure seem largely independent of the type of metal sample immersed.

**Table 5.** Molar contents of $NO_3{}^-$, $NO_2{}^-$ and $O_2{}^-$ in solar salt after 550 h of discontinuous corrosion testing at 600 °C in mol%. (a) Analysis by FZ Jülich, ZEA-3 and (b) double-analysis by DLR, Institute for Engineering Thermodynamics.

| Material in Salt | (a) | | | | (b) | | |
|---|---|---|---|---|---|---|---|
| | 550 h | | 1542–2042 h | | 550 h | | |
| | $NO_3{}^-$ | $NO_2{}^-$ | $NO_3{}^-$ | $NO_2{}^-$ | $NO_3{}^-$ | $NO_2{}^-$ | $O_2{}^-$ |
| Solar salt (initial) | 100.00 | <0.01 | 100.00 | <0.01 | 100.000 | <0.001 | 0.000 |
| Haynes 214 | 90.49 | 9.51 | 91.67 | 8.33 | 90.359 | 9.513 | 0.128 |
| Haynes 214 pre-ox. | 89.95 | 10.05 | 92.55 | 7.45 | 90.099 | 9.768 | 0.132 |
| LB2230 | 88.76 | 11.24 | 90.99 | 9.01 | 90.192 | 9.741 | 0.067 |
| LB2230 pre-ox. | 89.84 | 10.16 | 90.21 | 9.79 | 90.221 | 9.683 | 0.097 |
| Crofer®22 H | 89.22 | 10.78 | 92.15 | 7.85 | 90.333 | 9.563 | 0.104 |

### 3.4.2. Metal Dissolution

Table 6 summarizes the quantitative analysis results of metal alloying elements dissolving in the salt. Al, Ni, Fe, W, Nb, Si and Mn do not seem to migrate substantially from the metal samples into the solar salt (at least over durations of approximately 550 h, i.e., from 0 to 550 and 1542 to 2042; fresh salt was applied at 1542 h). The Cr content, instead, significantly increased depending on the exposed material and pre-treatment.

**Table 6.** Metal species dissolving in solar salt (ppm) over a duration of (a) 0 to 550 h and (b) 1542 to 2042 h (fresh salt applied at 1542 h) of corrosion testing at 600 °C.

| Material in Salt | (a) | | | | | | | | (b) |
|---|---|---|---|---|---|---|---|---|---|
| | Cr | Al | Ni | Fe | W | Nb | Si | Mn | Cr |
| Solar salt (initial) | 23 | <30 | 13 | 17 | 31 | <30 | <200 | 16 | 23 |
| Haynes 214 | 160 | <30 | 4 | <10 | <30 | <30 | <200 | 6 | 22 |
| Haynes 214 pre-ox. | 83 | <30 | <2 | <10 | <30 | <30 | <200 | 2 | 78 |
| LB2230 | 58 | <30 | <2 | 12 | <30 | <30 | <200 | 1 | 19 |
| LB2230 pre-ox. | 49 | <30 | <2 | <10 | <30 | <30 | <200 | <1 | 25 |
| Crofer®22 H | 135 | <30 | <2 | <10 | <30 | <30 | <200 | <1 | 52 |

After 550 h, the highest dissolved chromium content was encountered in the case of Haynes 214, followed by Crofer®22 H. Pre-oxidation of Haynes 214 approximately halved the amount of chromium in the salt, which indicates a protective effect of the mixed chromia/alumina pre-oxidation scale. At 1542 h, the crucibles were filled with fresh salt, i.e., the driving concentration gradient for metal species' diffusion was reset to the maximum. With the exception of pre-oxidized Haynes 214, less Cr dissolved into the salt from all the alloys over the following 500 h period from 1542 to 2042 h. In comparison to Haynes 214, the ferritic LB2230 alloy released the lowest quantity of Cr regardless of the surface condition prior to exposure. Pre-oxidation of LB2230 in air yielded a comparatively small further decrease by approximately 16%, which indicates the formation of a protective $Al_2O_3$ scale by exposure of the bare metal surface to molten solar salt.

### 4. Conclusions

An Al-alloyed, low-cost, ferritic, Laves phase-strengthened steel (LB2230) was benchmarked against the established Al-alloyed Haynes 214 Ni-base and the ferritic, $Cr_2O_3$-forming, Laves phase-strengthened stainless-steel Crofer®22 H, considering corrosion resistance and microstructure evolution during molten salt exposure. The takeaway messages of the presented practical research work can be summarized as follows:

- In air, temperatures typically higher than 1000 °C are needed to grow protective $Al_2O_3$ scales on the Al-alloyed trial steel LB2230 and the Ni-base alloy Haynes 214.
- Protective Al mixed oxide scales are obtainable at 600 °C in molten 60 wt.% $NaNO_3$– 40 wt.% $KNO_3$ solar salt on these materials.
- The Al-alloyed Ni-base superalloy Haynes 214, which was originally developed for application in molten salt environment, demonstrated good performance over a testing period of 2000 h.
- Uniform internal oxidation of Haynes 214 reached a depth of approximately 15 μm.
- The ferritic, stainless, Al-alloyed, Laves phase-strengthened trial steel LB2230 demonstrated weight gains on the niveau of Haynes 214 in discontinous and isothermal salt corrosion experiments.
- Sporadic internal oxidation of LB2230 was limited to a depth of approximately 5 μm.
- LB2230 released a minimum of Cr species into the molten salt.
- With the exception of minimized initial Cr-release in case of Haynes 214, pre-oxidation of the Al-alloyed materials in air did not yield significantly different results.
- The ferritic, $Cr_2O_3$-forming, Laves phase-strengthened Crofer®22 H steel presented the highest weight changes, did not form stable, protective oxide scales and tended to spallation, especially in discontinous corrosion testing. Therefore, this alloy seems unsuitable for long-term application in molten solar salt.
- In comparison to Haynes 214, decreased internal oxidation, reduced dissolution of Cr species into the molten salt and low material costs make LB2230 a favorable candidate alloy for CSP and TES application.
- $NO_3$ dissociation from solar salt depends on the formation of N-consuming oxide/precipitate phases on/in the immersed metals.
- The strenthening Laves phase precipitates in the ferritic steels were demonstrated to be stable during exposure to molten salt. The strengthening by Laves phase particles in combination with potential self-passivation in case of the $Al_2O_3$-forming LB2230 alloy poses immense cost-saving potential over austenitic stainless-steel and especially Ni-base superalloys.

Materials for salt guiding pipework and containers in future CSP and TES facilities should ideally be self-passivating in service. On the other hand, passivating oxide scales could be applied during material or component manufacturing (for example, during mandatory recrystallization/solution heat treatment of sheet material, pipes and tubes) without additional cost. However, especially in case of weldments, where downstream heat treatment possibilties are limited or even prohibited, in-service self-passivation would be a real game changer. Commercial Al-alloyed, and on the basis of $Al_2O_3$-formation, potentially self-passivating, materials are either prohibitively expensive (Haynes 214, Haynes 224) or mechanically weak at operation temperatures of $\geq$600 °C (commercial Al-alloyed ferritic steels [25,39,40,66]).

To conclude the presented results, in a technological context, it can be stated that ferritic, stainless, Al-alloyed, Laves phase-strengthened LB2230-like steels may provide a possible way out of this dilemma. New steels, based on this alloying philosophy, provide viable molten salt corrosion resistance by self-passivation, mechanical high-temperature strength far superior to commercial FeCrAl steels, in combination with feasably low material costs (approximately factor 20 in comparison to Ni-base/factor 5 in comparison to austenitic steel).

Nevertheless, ongoing research needs to focus on the following issues:

- Increased exposure times to molten salt to safeguard corrosion resistance and microstructural stability.
- In-depth research on corrosion mechanisms and potential competition between oxide scale formation and Laves phase stability in near-surface regions.
- Mechanical property evaluation in molten salt environment.
- Detailed examinatoin of self-healing capacity of cracked oxide scales, especially under thermomechanical fatigue conditions.

**Author Contributions:** Conceptualization, F.A. and B.K.; methodology, F.A., B.K., A.B. and T.B.; investigation, F.A. and A.B.; resources, B.K., A.B. and T.B.; data curation, F.A. and A.B.; writing—original draft preparation, F.A.; writing—review and editing, B.K., F.A., A.B. and T.B.; visualization, F.A. and B.K.; supervision, B.K., A.B. and T.B.; project administration, B.K., A.B. and T.B.; funding acquisition, B.K., A.B. and T.B. All authors have read and agreed to the published version of the manuscript.

**Funding:** Development and production of the LB2230 trial steel was funded by the German Federal Ministry of Education and Research (BMBF) under grant number 03x3520E, which is greatly appreciated. Supply of Haynes 214 materials by Haynes International, Switzerland, and Zapp Materials Engineering GmbH, Germany, is acknowledged.

**Institutional Review Board Statement:** Not applicable.

**Informed Consent Statement:** Not applicable.

**Data Availability Statement:** Data sharing is not applicable.

**Acknowledgments:** The authors also wish to thank the following staff members of Forschungszentrum Juelich GmbH: E. Wessel and D. Grüner (microstructural examination).

**Conflicts of Interest:** The authors declare no conflict of interest. The funders had no role in the design of the study; in the collection, analyses, or interpretation of data; in the writing of the manuscript, or in the decision to publish the results.

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
