# Peer review of "A New Approach to Low-Cost, Solar Salt-Resistant Structural Materials for Concentrating Solar Power (CSP) and Thermal Energy Storage (TES)"

_metals, doi:10.3390/met11121970_

Round 1
Reviewer 1 Report
Dear Authors,
The paper explained A New Approach to Low-cost, Solar Salt Resistant Structural Materials for Concentrating Solar Power (CSP) and Thermal.
Energy Storage (TES). It's well organized; however, the following comment may need to match the paper for publishing in the Journal:
1) A lack of recent works are clear in the introduction, while more than 70% of references are older than 2016. Please use some recent research.
2) Is there any concept behind using three specific structural materials? Please explain in section 2.1.1.
3) Figure 4 and5: Please use higher resolution photos.
4) Please add the histogram for EDX analysis which gives a better overview of chemical concentration.
5) The approach used by the authors is acceptable, but I can't see in the conclusion what is the primary outcome. 3 alloys compared; however not clear which one considering the conditioning is the best. Highly recommend revising the conclusion to give a clear outcome for the readers.
6) Englis need to be checked by a native technical English speaker.
Author Response
1) Where it fit, I have added newer references or replaced older references with newer ones. In this way I was able to raise the annual average: I have replayed:- Müller-Steinhagen H.; Trieb. F. 2004 with Liu, M. et. al. Review on concentrating solar power plants and new developments in high temperature thermal energy storage technologies 2016
- Geyer M.; et. al. Dispatchable Solar Electricity for Summerly Peak Loads from the Solar Thermal Projects Andasol 1 and Andasol 2 2006 with Siefert J.A.; et. al. Concentrating solar power (CSP) power cycle improvements through application of advanced materials 2015
- Adams D. Efficiency upgrades and partial carbon capture for coal fired power plants 2009 with Zhang T. Methods of Improving the Efficiency of Thermal Power Plants 2020
- Pfleger N.; et. al. Material aspects of solar salt for sensible heat storage 2013 with Ding, W.; et. al. Development of molten chloride salts for thermal energy storage in next generation concentrated solar power (CSP) plants 2018
- Bauer, T. et. al. Molten Salt Storage for Power Generation 2021
- Mallco Carpio, A. et. al. Corrosion and Mechanical Assessment in LiNO3 Molten Salt as Thermal Energy Storage Material in CSP Plants 2019
- Garcia-Martin G.; et. al. Corrosion Behavior of VM12-SHC Steel in Contact with Solar Salt and Ternary Molten Salt in Accelerated Fluid Conditions 2021
In the context of this work, two ferritic stainless steels, strengthened by Laves phase precipitates, were investigated: Crofer®22 H, a chromia former and LB2230, an experimental alumina forming derivate of Crofer®22 H. The high chromium content of Crofer®22 H provides excellent steam oxidation and corrosion resistance [26]. High creep resistance of this type of steel is achieved by combined solid solution and intermetallic Laves particle precipitation strengthening [26]. This in principle makes the steel conceivable for structural CSP power plant application, with long-term resistance to molten salt corrosion remaining a point of concern. LB2230, an α-Al2O3 forming, Laves phase strengthened derivate of Crofer®22 H, potentially provides significantly better molten salt corrosion resistance compared to its chromia forming forerunner Crofer®22 H [27]. The performance of these two low-cost ferritic grades is compared to the alumina forming Ni-base super alloy Haynes 214, which was developed for application in molten salt and for this reason serves as a benchmark.
3) I have replaced the images with higher resolution images.4) I asked our SEM lab for histograms and looked at them in the context of the results. Adding these makes little sense in my eyes. They fill the paper and have only little significance in comparison. Nevertheless, I understand the desire for a better overview of the chemical compositions and have therefore added a sum spectrum in each case. 5) We have revised part of the conclusion to present the results more clearly:
- The ferritic, Cr2O3-forming, Laves phase strengthened Crofer®22 H steel presented the highest weight changes, did not form stable, protective oxide scales and tended to spallation especially in discontinous corrosion testing. Therefore, this alloy seems unsuitable for long-term application in molten solar salt.
- In comparison to Haynes 214 decreased internal oxidation,reduced dissolution of Cr species into the molten salt and low material costs make LB2230 a favorable candidate alloy for CSP and TES application.
- NO3—dissociation from solar salt depends on the formation of N-consuming oxide / precipitate phases on / in the immersed metals.
- The strenthening Laves phase precipitates in the ferritic steels demonstrated to be stable during exposure to molten salt. The strengthening by Laves phase particles in combination with potential self-passivation in case of the Al2O3-forming LB2230 alloy poses immense cost-saving potential over austenitic stainless steel and especially Ni-base superalloys.
Reviewer 2 Report
First thing first, this is a decent paper and is possible for publication in the present journal. It provides a new design route for corrosion (solar salt)-resistant stainless steels. From my perspective, moderate revisions are required for its chance of getting published.
Major:
1. It was not until I read through the whole article to the last paragraphs, did I know that LB2330 (or LB2330-like) was the one that the authors wanted to promote. I had a hard to distinguish which one was better among the others and follow the related main messages. And some words are misleading, e.g. on Page 2 'The performance of these is benchmarked to the alumina forming Ni-base super alloy Haynes 214, which is known for superior resistance to solar salt' which seems that Haynes 214 is the one that's promoted but actually not. I am sure the general audience will get stuck on this as I did. Therefore, please add some heads-up for LB2330 either in the ending area of the introduction section or some other place that you think is reasonable.
2. This is a general issue when I review experiment-heavy papers, that often times the sciences behind these measurements are buried. You plotted mass gains, and oxide scale and Laves phase micrographs, but the sciences or the mechanisms behind them are not well discussed, e.g. why the oxide scales seem more stable in LB2330 and why the Laves phase formation is hindered near the surface of LB2330 (I'm not asking you to answer exactly what I'm asking). The authors need to come into a few conclusive words (or paragraph) saying these are why LB2330 is better and we should choose it. This is crucial to help the reader catch up the sciences and main messages.
In-between major and minor:
1. On the top of Page 9, the present experiment seems to contradict the previous one in terms of the protective mechanism in Haynes 214. It is necessary to add more discussions for the reasons, e.g. different experimental conditions?
Minor:
1. What is 'Ma.-%' on Page 5?
2. On Page 10, it should be Table 3 not Table 1.
3. On Page 13, 'giuding' should be 'guiding'.
Author Response
The answers to your comments are shown in red
1. It was not until I read through the whole article to the last paragraphs, did I know that LB2330 (or LB2330-like) was the one that the authors wanted to promote. I had a hard to distinguish which one was better among the others and follow the related main messages. And some words are misleading, e.g. on Page 2 'The performance of these is benchmarked to the alumina forming Ni-base super alloy Haynes 214, which is known for superior resistance to solar salt' which seems that Haynes 214 is the one that's promoted but actually not.
Response to reviewer: The statement has been changed to: "The performance of these two low-cost ferritic grades is compared to the alumina forming Ni-base super alloy Haynes 214, which was developed for application in molten salt and for this reason serves as a benchmark."
I am sure the general audience will get stuck on this as I did. Therefore, please add some heads-up for LB2330 either in the ending area of the introduction section or some other place that you think is reasonable.
Response to reviewer: Sentence 4 of the abstract hase been modified to: "This paper outlines the superior salt corrosion behavior of a novel low-cost, Al2O3-forming, ferritic, Laves phase strengthened (i.e. structural) steel in NaNO3/KNO3 solar salt at 600 °C." The statement "This paper outlines the performance of a new type of Laves phase strengthened, low-cost, salt corrosion resistant ferritic steel (called “LB 2230”, cf. section 2)" has been added at the end of the introduction.2. This is a general issue when I review experiment-heavy papers, that often times the sciences behind these measurements are buried. You plotted mass gains, and oxide scale and Laves phase micrographs, but the sciences or the mechanisms behind them are not well discussed, e.g. why the oxide scales seem more stable in LB2330
Response to reviewer: Some modifications have been done at paragraph 1 of section 3.3, outlining that Haynes214 presents internal oxidation, what poses doubts concerning long-term protection. The statements "Scale cracking did not occur in case of LB2230, which is presumably caused by the lower mismatch of thermal expansion coefficients of the formed oxide scales and the ferritic matrix."… and … "In contrast to Haynes 214 internal oxidation of the LB2230 substrate material beneath the ??2?3 layer does not occur, what indicates higher stability of the protective layer. " have been added to the 3rd paragrapg of section 3.3.and why the Laves phase formation is hindered near the surface of LB2330 (I'm not asking you to answer exactly what I'm asking). The authors need to come into a few conclusive words (or paragraph) saying these are why LB2330 is better and we should choose it. This is crucial to help the reader catch up the sciences and main messages.
Response to reviewer: The issue of differing Laves phase precipitation in the surface near regions is discussed in detail in section 3.3.1. Furthermore the statement "Homogenous precipitate microstructure, like in case of LB2230, is desirable, because it provides maximized strength at the same material cross section and thus enables thinner section components." has been added to outline that LB2230 is favorable.In-between major and minor:
1. On the top of Page 9, the present experiment seems to contradict the previous one in terms of the protective mechanism in Haynes 214. It is necessary to add more discussions for the reasons, e.g. different experimental conditions?
Unfortunately, I did not understand what you meant by this. Section 2.2 shows the experimental methods and the two differences between the isothermal experiment at DLR and the thermocyclic experiment at FZJ. There were no other experiments with different experimental conditions within the manuscript.
Minor:
1. What is 'Ma.-%' on Page 5?
2. On Page 10, it should be Table 3 not Table 1.
3. On Page 13, 'giuding' should be 'guiding'.
I have corrected the points in the manuscript.
Reviewer 3 Report
Comment 1: Qualitative informations are missing in abstract. Abstract should be concise and the authors need to improve with more specific short results.
Comment 2: The introduction section should be modified though citing recent references related studies and indicating the novelty of the study compared to the carried works.
Comment 3: The following references should be added.
Composite Structures 262 (2021) 113640 https://doi.org/10.1016/j.compstruct.2021.113640
Comment 4: Level of English is good however in a few places some syntax errors are present. At some places two or more words joined together that should be corrected.
Comment 5: Compare your results with literature ones.
Author Response
Comment 1: Qualitative informations are missing in abstract. Abstract should be concise and the authors need to improve with more specific short results.
The abstract states the result of the three materials studied, which of them performs best.
Comment 2: The introduction section should be modified though citing recent references related studies and indicating the novelty of the study compared to the carried works.
The abstract states the result of the three materials studied, which of them performs best. It also states what the study was done for. A more detailed summary of the results makes more sense after you have read the main part. In the Conclusion, I have added a few more things to present the results more clearly.
Comment 3: The following references should be added.
Composite Structures 262 (2021) 113640 https://doi.org/10.1016/j.compstruct.2021.113640
I looked at the reference and quickly realised that it did not fit thematically. Probably the review was addressed to another paper, or I should actually include another reference.
Comment 4: Level of English is good however in a few places some syntax errors are present. At some places two or more words joined together that should be corrected.
I read through the manuscript again and was able to correct 2 or 3 linguistic errors. In agreement with the co-authors, the language level is appropriate.
Comment 5: Compare your results with literature ones.
Where possible, the results have already been compared with results from the literature (e.g. Haynes 214). LB2230 and Crofer22 H have not yet been studied in solar salt, or there is no literature on them.
Round 2
Reviewer 2 Report
Thanks for these responses to my comments. For the last major comment where you did not understand what I had meant, I've actually saw you modified that part of paragraph and it reads fine. Therefore, all the comments are properly addressed.